# Hypofractionated Radiotherapy in Locally Advanced Myxoid Liposarcomas of Extremities or Trunk Wall: Results of a Single-Arm Prospective Clinical Trial

**DOI:** 10.3390/jcm9082471

**Published:** 2020-08-01

**Authors:** Hanna Koseła-Paterczyk, Mateusz Spałek, Aneta Borkowska, Paweł Teterycz, Michał Wągrodzki, Anna Szumera-Ciećkiewicz, Tadeusz Morysiński, Patrycja Castaneda-Wysocka, Andrzej Cieszanowski, Marcin Zdzienicki, Tomasz Goryń, Piotr Rutkowski

**Affiliations:** 1Department of Soft Tissue/Bone Sarcoma and Melanoma, Maria Sklodowska-Curie National Research Institute of Oncology, 02-781 Warsaw, Poland; Mateusz.Spalek@pib-nio.pl (M.S.); Aneta.Borkowska@pib-nio.pl (A.B.); Pawel.Teterycz@pib-nio.pl (P.T.); tmorysinski@coi.waw.pl (T.M.); Marcin.Zdzienicki@pib-nio.pl (M.Z.); Tomasz.Goryn@pib-nio.pl (T.G.); Piotr.Rutkowski@pib-nio.pl (P.R.); 2Department of Pathology and Laboratory Medicine, Maria Sklodowska-Curie National Research Institute of Oncology, 02-781 Warsaw, Poland; Michal.Wagrodzki@pib-nio.pl (M.W.); Anna.Szumera-Cieckiewicz@pib-nio.pl (A.S.-C.); 3Diagnostic Hematology Department, Institute of Hematology and Transfusion Medicine, 02-781 Warsaw, Poland; 4Department of Radiology, Maria Sklodowska-Curie National Research Institute of Oncology, 02-781 Warsaw, Poland; p.castaned@gmail.com (P.C.-W.); Andrzej.Cieszanowski@pib-nio.pl (A.C.)

**Keywords:** myxoid liposarcoma, hypofractionated radiotherapy, neoadjuvant

## Abstract

Introduction: Myxoid liposarcoma (MLPS) has been reported to be more radiosensitive compared with other soft tissue sarcomas. The main objective of the study was to assess the efficacy of hypofractionated radiotherapy (RT) in the preoperative setting in patients with locally advanced primary MLPS. Methods: Single-arm prospective exploratory clinical trial enrolled MLPS patients for preoperative 5 × 5 Gy RT with delayed surgery. The endpoints of the study were the rate of early wound healing complications and 5-year local control rate. Results: 29 patients (pts) were included, all had tumors located on the lower limb. The median maximum size of the tumor was 13 cm (IQR 10–15 cm). Early RT tolerance was good. Postoperative wound complications occurred in 11 pts (37.9%), late complications concerned 13.8% of patients. A total of 27 patients were included for the efficacy analyses. The pathological features of response to RT were detected in all analyzed surgical specimens. In 25 patients R0 margins were achieved, two patients had an R1 resection. None of the patients had local recurrence. Conclusion: Preoperative hypofractionated RT with a prolonged gap between RT and surgery is a feasible method of the management of MLPS, providing a good local control and low rates of treatment toxicity.

## 1. Introduction

Liposarcomas are one of the most common soft tissue sarcomas (STS). The myxoid liposarcoma (MLPS) subtype represents approximately 30% of all liposarcomas with peak incidence in the fourth decade of life. Most often, MLPS is localized in the deep soft tissue of the extremities, usually the proximal thigh [1,2]. Pathological findings show a lesion composed of uniform, round, or oval primitive nonlipogenic mesenchymal cells with lipoblasts gathered peripherally or near the blood vessels, surrounded by myxoid stroma [3]. Presence of round cell component and its percentage is a well-known poor prognostic factor. Diagnosis of MLPS can be confirmed by the finding of a specific chromosomal translocation t(12;16)(q13;p11), detected in more than 95% of cases, which results in fusion of *DDIT3* (*CHOP*) gene and *FUS* (*TLS*) gene [4,5].

The primary treatment of locally advanced STS, including MLPS, is a wide surgical resection of the tumor. For the majority of patients with high/intermediate grade and many low grade STS, the use of adjuvant radiotherapy (RT) should be considered in order to improve the local control of the disease [6]. Randomized phase III clinical trials comparing preoperative to postoperative RT showed comparable results in terms of efficacy with more acute toxicity of the preoperative schedule and more long-term toxicity, e.g., fibrosis in the postoperative scheme [7,8]. The recommended fractionation regimen of preoperative RT is 50–50.4 Gy in 25–28 fractions (1.8–2.0 Gy per fraction) with 5–6 weeks break between the RT ending and surgery [6,9]. Our previous study showed that in patients with locally advanced STS, hypofractionated preoperative RT provided a good local control and manageable toxicity [10].

MLPS, when compared to other STS subtypes, shows higher sensitivity to RT. There are reports of the substantial decrease in the volume of lesions and also histological features of response to radiation treatment such as induction of adipogenesis [11,12]. Furthermore, in MLPS patients treated with the aforementioned hypofractionated RT regimen, pathological response to therapy was observed despite the very short gap between the RT completion and surgery, i.e., between two and seven days [13].

The aim of this study was to assess the efficacy and safety of preoperative hypofractionated RT followed by delayed surgery in patients with locally advanced MLPS.

## 2. Materials and Methods

We conducted a phase II nonrandomized prospective exploratory clinical trial (NCT03816475). Ethical approval for this study was obtained from the ethics committee of Maria Sklodowska-Curie National Research Institute of Oncology in April 2015, code number 12/2015. In the study, we included patients with primary, locally advanced myxoid liposarcomas located on the limbs or trunk, sized equal, or more than 5 cm in greatest dimension. All patients were 18 or older and had Eastern Cooperative Oncology Group (ECOG) performance status 0 to 2. Exclusion criteria included evidence of distant metastases, planned neoadjuvant or adjuvant chemotherapy, prior RT to the area to be irradiated, and active treatment of a second malignancy. All MLPS diagnoses were centrally reviewed in our center by diagnostic pathologists experienced in soft tissue sarcomas and confirmed by molecular testing for *DDIT3* rearrangements According to NCCN and ESMO guidelines, preoperative RT is not a routine treatment in patients with stage I sarcomas. However, it may be considered if we expect achieving oncologically inappropriate margins, especially in larger tumors (stage IB). Such a consideration was discussed during sarcoma multidisciplinary tumor board in each patient with locally advanced MLPS. We assumed a good response to preoperative RT that would lead to tumor downstaging as well as to elimination of microscopical infiltration of sarcoma cells around the primary tumor.

The primary endpoint of the study was the rate of serious wound complications defined similarly to assumptions taken in the randomized phase III clinical trial conducted by the National Cancer Institute of Canada [7] and assessed by National Cancer Institute Common Terminology Criteria for Adverse Events, version 4.0. (CTCAE v.4) which means: A necessity for a secondary operation under general or regional anesthesia for wound repair (debridement, operative drainage, and secondary wound closure including rotationplasty, free flaps, or skin grafts)—Grade 3 according to CTCAE v.4Wound infection requiring systemic antibiotics—Grade 3 according to CTC AE v 4.Persistent deep packing for 90 days or longer (due to wound dehiscence)—Grade 2 according to CTCAE v4.Hospital readmission for wound care.

To assess the primary endpoint, patients were followed up at least once a week for the first month after surgery, every second week for the second month, and monthly thereafter for 3 months. The assumption of the study was that it would be terminated earlier in the case of unacceptable toxicity of treatment defined as the frequency of occurrence of toxicity ≥grade 2 according to CTCAE in over 40% of the treated patients. The estimated number of patients to be included, based on this exploratory primary endpoint, was 25–30.

The secondary endpoints were pathological response to RT, radiological response in magnetic resonance (MRI), and local disease control at three and five years. Each participant provided written informed consent. In all patients, before treatment the tumor staging was performed by physical examination, MRI and computed tomography (CT) imaging of the primary tumor and a contrast-enhanced CT of the chest, abdomen, and pelvis. After treatment completion, all patients were followed for evidence of local recurrence or distant metastases. Patients were assessed every 3 months in the first 2 years, then twice a year up to the fifth year, and once a year thereafter. Alternate ultrasound (US) and MRI of the scar were performed, as well as CT of the chest abdomen and pelvis, alternating with X-ray of the chest and US of the abdomen.

The recruitment commenced in April 2015 and lasted for 48 months. Twenty-nine patients were included in the trial and two of them are excluded from the final efficacy analyses, as the diagnosis of MLPS was not confirmed in the final assessment of postoperative specimen. All patients treated in the protocol (intent-to-treat cohort) were included in the safety analysis.

All tumors were located on the lower limbs. Median tumor diameter was 13 cm (range 5–20 cm). Patient characteristics are shown in Table 1. Detailed patient characteristics are shown in Appendix A.

### 2.1. Assessment

Radiological response was assessed according to the Response Evaluation Criteria in Solid Tumors (RECIST), version 1.1. The severity of adverse events was graded according to the CTCAE v.4 criteria. Safety monitoring was based on the occurrence of early toxicity.

### 2.2. Treatment

After individually chosen immobilization and planning CT, the delineation process was performed by an experienced radiation oncologist. Contours were independently reviewed by another radiation oncologist. The gross target volume (GTV) was contoured on planning CT fused with contrast-enhanced MRI and/or diagnostic CT. Clinical target volume (CTV) was created by expanding GTV 2 cm transversally and 4 cm longitudinally and reduced at anatomical borders of the tumor spreading (i.e., bones and fascias), unless involved. Planning target volume (PTV) was created by expanding CTV, adding safety margins (0.7–1 cm). Three-dimensional RT techniques with or without dose intensity modulation with daily image guidance (planar kilovoltage or cone-beam computed tomography) were used to deliver the prescribed dose. Wide local excision was performed after a median of 7 weeks (range 5–10 weeks) from the end of RT. The tissue flaps were not used.

### 2.3. Patient Samples

Pathological samples from all patients were reviewed by two pathologists (MW, AS-C). Tissue samples (both biopsy material and post-surgical specimens) were fixed in 10% neutral buffered formalin, routinely processed, embedded in paraffin, and stained with hematoxylin and eosin. The percentage of round cell component (RCC) was evaluated and defined as positive when the RCC comprised more than 5% of tumor cells. The cases were divided into low and high histological grades on biopsy evaluation. Previously described cutoff point with >5% RCC or the presence of tumoral necrosis regardless of the RCC define the high-grade MLPS according to FNCLCC grading system [5,14,15]. Fluorescence in situ hybridization (FISH) was performed to confirm the rearrangements of *DDIT3* gene in every case. In post-treatment surgical material, the percentage of RCC, percentage of necrosis, and other RT histological response features (viz. hyalinization, fibrosis, paucicellularity, hemorrhages, dilatation of vessels, adipogenesis induction) were described. Response to treatment was also assessed by the European Organization for Research and Treatment of Cancer-Soft Tissue and Bone Sarcoma Group (EORTC-STBSG) recommendations for pathological examination and reporting [16]. Photographs were taken using the SC50 5-megapixel color microscope camera (Olympus, Japan). The pathological characteristics of the patients are shown in Table 1.

### 2.4. Statistical Analysis

All analyses were performed in the R language environment version 3.5.1 (The R Foundation for Statistical Computing, Vienna, Austria). Patient demographics, tumor characteristics, and details of treatment and tumor response were analyzed in a descriptive fashion. The median follow up for the surviving patients was 27.1 months (IQR 20.2–35.7). The local recurrence-free survival (LRFS), disease-free survival (DFS), and overall survival (OS) were estimated according to the Kaplan–Meier method. LRFS time was calculated from the date of the start of the preoperative RT to the date of the most recent follow-up (censored) or local recurrence. DFS time was calculated from the date of the start of preoperative RT to the date of the most recent follow-up (censored) or recurrence. OS time was calculated from the start of RT to the most recent follow-up (censored), or death.

## 3. Results

All 29 patients completed the scheduled therapy without unplanned breaks in treatment. Eighteen patients were irradiated with three-dimensional conformal RT, 6 patients with intensity-modulated RT and 5 patients with volumetric modulated arc therapy. Median GTVs, CTVs and PTVs were as follows: 389 cm^3^ (IQR: 249–836), 1832 cm^3^ (IQR: 1176–2653), 2605 cm3 (IQR: 1772–3621 cm^3^).

As previously noted, 27 patients were included in the efficacy analyses. All but two patients had an R0 resection of the tumor, two patients (7.4%) had an R1 resection. None of the patients had local recurrence of the disease. During follow-up in 4 patients (14.8%), metastatic disease was diagnosed, and all of these patients had high-grade tumors. At the time of the analysis two patients had died due to the distant spread of the sarcoma. The only factors having a significant impact on the DFS were the grade (*p* = 0.00074, by log-rank test) (Figure 1) of the tumor and the percentage of RCC (*p* = 0.009; HR = 1.270 per 1% change; 95%CI 1.063–1.518).

Twelve patients had radiological responses also assessed by RECIST criteria by comparing scans before surgery to those done prior to treatment. In 7 (58.3%), partial response to treatment was achieved, the remaining 5 (41.6%) had stable disease. Changes in the density of the lesions after treatment were also noted (Figure 2).

The rearrangements in the *DDIT3* gene were confirmed by the FISH technique in all 27 patients included in the final efficacy analysis. In all analyzed surgical specimens, RT histological response features (i.e., hyalinization, fibrosis, necrosis) were detected. In all but one patient, adipogenesis induction was noted as a response to therapy. In 8 patients (30%), adipogenesis was seen in more than 50% of the postoperative specimen (Figure 3). In 1 patient (3.7%), response to treatment was noted as a according to EORTC-STBSG response score, in 0 as B, in 8 (29.6%) as C, in 10 (37%) as D and in the remaining 8 (29.6%) as E.

### Toxicity

All patients (29) who signed the informed consent were included in the safety analyses. Treatment was well tolerated with a low rate of late morbidity. The study met the primary endpoint. Radiation dermatitis was assessed within one week prior to the surgery, 17 (58.6%) had no signs of radiodermatitis, 10 (34.4%) had grade 1 radiodermatitis, one had grade 2 and one had grade 3 radiodermatitis (3.4%). Eleven patients (37.9%) experienced early treatment-related toxicity, with prolonged wound healing grade 1 as the most common. Eight patients (27.5%) had more severe wound complications; in 3 (10.3%) it was wound dehiscence (grade 2) and in 5 (17.2%) it was wound infection requiring oral antibiotic use (grade 3). None of the patients had significant wound healing problems necessitating additional surgical intervention.

Late toxicities of the treatment were noted in 4 patients (13.8%). In two patients this was limb edema, grade 1 in one patient and grade 2 in the other. Fibrosis of the soft tissue was also found in two patients, grade 1 in one patient and grade 2 in the second. The patient with grade 2 fibrosis also experienced grade 1 paresthesia. No grade 3 or higher toxicities were observed after a median follow-up of 27 months (minimum 11 months). The summary of the treatment toxicities is shown in Table 2.

## 4. Discussion

Our study showed that hypofractionated RT is efficient, safe, and feasible in patients with MLPS who are qualified for the preoperative treatment. Two randomized prospective clinical trials published more than twenty years ago showed that the addition of adjuvant radiation (brachytherapy or external beam) therapy allows decreasing of the amount of local recurrence after conservative, limb-sparing surgery [17,18]. Combined therapy is now standard in the recommendations for the treatment of majority of patients with STS [6,19]. RT in STS can be used before or after surgery. The randomized phase III clinical trial conducted by the National Cancer Institute of Canada showed that both sequences of treatment appear to give comparable results in terms of local control of the disease. Both methods vary in terms of complication rates. Preoperative treatment is associated with more early wound complications (35% vs. 17%), while postoperative radiation therapy has a higher risk of late side effects such as lymphedema, fibrosis, or scarring [7,8].

The undoubtable benefit of using preoperative RT is the chance to reduce the tumor size, obtain thickening of the tumor pseudocapsule, and thus increase the chance of radical resection in the case of large borderline resectable lesions [20]. This can especially be seen in STS subtypes sensitive to radiation, such as MLPS. Pitson et al. compared the changes in tumor volume after irradiation of patients with MLPS and malignant fibrous histiocytomas (MFH), now called undifferentiated pleomorphic sarcomas (UPS). The proportional reduction in the median volume of the tumor was 59% for MLPS and −7% for MFH tumors [21]. Furthermore, a Dutch study showed that within a group of various analyzed STS subtypes, MLPS was the one with the most pronounced decrease in the tumor volume after radiation therapy [12]. In our group, a significant response rate in terms of tumor size was also observed, with 58% PR to treatment in the assessed patients according to RECIST criteria. None had progressive disease. The assessment of response to therapy has some shortcomings as we decided to use the RECIST criteria, which is still a validated system for response evaluation in clinical trials. As previously mentioned, it has been shown that myxoid liposarcomas often respond to treatment with a significant size shrinkage, but we did not assess in a planned manner the changes in the density of the lesions. Choi criteria are used more commonly in the assessment of treatment in sarcomas [22] with some evidence that those criteria can be more adequate than RECIST in predicting the patient outcome [23]. We plan to retrospectively assess those criteria in our patients group and in our future studies we plan on using both RECIST and the Choi criteria.

A unique response to RT in MLPS can also be noted in the pathological examination of the surgical samples. Except for a high proportion of classical post-treatment changes such as necrosis or fibrosis, which were noted in all of the treated patients, a striking maturation of tumor cells to lipocytes can also be observed. In all but one patient in our group, adipogenesis induction as a response to therapy was detected. In 8 patients (30%), adipogenesis was seen in more than 50% of the postoperative specimen. This remarkably interesting effect of treatment was also observed by Enström et al. who detected new (in comparison with preoperative samples) lipomatous differentiation in 21 out of 27 MLPS [11]. Similar effects were found after the use of trabectedin in preoperative therapy, a drug that shows distinctive efficacy in the treatment of MLPS [24]. A total of 29 patients with locally advanced MPLS were treated with neoadjuvant three to six cycles of trabectedin, achieving a 24% response rate. Three patients had a complete pathological response to therapy, and the specimens in the treated population also showed maturation of the tumor cells into lipocytes, along with the reduction of cellularity and vasculature of the lesions [25]. Adipogenesis should be further evaluated in comparison with other standard histological response features in terms of its prognostic significance in MLPS after treatment.

Based on the above observations, a phase I clinical trial was conducted in 14 patients with MLPS where both trabectedin and RT were combined in the preoperative treatment of localized disease. Chemotherapy was given in escalating doses for three cycles, and RT to a total dose of 45 Gy in 25 fractions (1.8 Gy/fraction), which is a smaller dose than usually used in the preoperative RT. The scheme was well tolerated, and the results are very promising: 36% of patients achieved PR by RECIST criteria, while 86% achieved PR by Choi criteria, assessing the changes in the density of the tumor. The median viable residual tumor in the postoperative specimen was 5%. A phase two study in MLPS with this combination is ongoing [26]. The results of the mentioned studies highlight the unique sensitivity of MLPS to RT, which also manifests itself in excellent local control of the disease. An analysis of a large database coming from a Canadian group showed that in a group of 691 STS patients treated with adjuvant RT, the population of 88 patients with MLPS achieved the best local control of the disease. The 5-year local recurrence-free survival was 97.7% for patients with myxoid liposarcoma compared with 89.6% for patients with other STS tumors (*p* = 0.008) [27]. In our group with a follow-up of median 27 months (minimum 11 months), none of the patients had a local recurrence of the disease.

Clinical studies are currently underway trying to address the question if the traditional fractionation and dose of RT in MLPS can be changed in order to minimize the side effects of the treatment, further improve the patient outcomes or improve the treatment convenience both for the patients and for the treating team [28,29]. We decided to explore the safety and efficacy of hypofractionated RT in the preoperative treatment of the patient with STS.

The advantages of hypofractionated RT include shorter overall treatment time, convenience for the radiation oncology team and patients, better reproducibility (fewer fractions), and probably a higher cost-effectiveness ratio. Shorter RT schemes can shorten time to treatment and increase the access to care at high-volume centers, which is especially important in the treatment for STS. An additional argument in favor of treatment in specialized centers comes among others from the Netherlands, where results of treatment of more than 5000 patients with STS were investigated. It was shown that surgery in a high-volume hospital (≥20 STS resection per year) improved the survival of patients with non-low-grade and deep-seated tumors. Specifically, 10-year survival rates for this group of patients were 54% for high-volume hospitals, and 49% and 42% for the low and medium volume hospitals, respectively, with a relative risk of 1.3 (high-volume versus low-volume, *p* = 0.03) [30]. Our center is the only one in Poland dedicated to the treatment of patients with soft tissue sarcomas, so shortening of the waiting list for treatment is a challenging and significant problem.

The main concern in the use of hypofractionation is a theoretically higher risk of late complications; however, it has not been well documented in hypofractionated regimens in other cancers. Much evidence for efficacy and safety of hypofractionation comes from prostate cancer where different schemes of fractionation have been developed and examined. Hypofractionated RT showed non-inferior results to conventionally fractionated RT for intermediate-to-high risk prostate cancer regarding failure-free survival. Late toxicity was similar in both treatment groups [31]. Similarly, the hypofractionated RT schemes have been profoundly studied in rectal cancer. No differences in oncological outcomes were reported between chemoradiotherapy with standard fractionation and surgery 4–6 weeks later and hypofractionated RT 5 × 5 Gy with surgery within 7 days, and there are no statistically significant differences in postoperative complications [32]. Short course preoperative RT with delayed surgery, a scheme similar to our study, is also a useful alternative to conventional short-course RT, with immediate surgery offering similar oncological outcomes and lower postoperative complications in patients with locally advanced rectal cancer [33].

We also published our experience with the use of hypofractionated RT 5 × 5 Gy with immediate surgery in a group of unselected patients with locally advanced STS of the limbs and trunk wall. The rate of late complications was 12.4%, which is comparable with reports coming from other RT schemes used in STS [10]. Even a higher dose of hypofractionated RT 30 Gy 5 times 6 Gy was used in the preoperative treatment of different subtypes of STS published recently by authors from the University of California. The rate of grade ≥2 radiation-associated toxicity (fibrosis, joint stiffness, or lymphedema) after median follow-up of two years was tolerable (16%). The oncology outcomes are good with a 5.7% rate of local recurrence after surgery [34]. In our current study the rate of late morbidity was also relatively low, with 13.8% of long-term radiation-induced toxicity.

There are also limitations of our study. Firstly, we did not use perioperative chemotherapy. Four patients developed metastatic disease and all of them had high grade tumors with more than 5% of the RCC component, what is a known factor impacting the survival of the patients [5,35]. Especially important in this aspect is the recently published study by Gronchi et al. comparing standard (epirubicin and ifosfamide) chemotherapy to histotype tailored. Though the results of the trial are formally negative (tailored chemotherapy showed no benefit over the standard one), it showed a favorable outcome for patients treated with epirubicin and ifosfamide, confirming the efficacy of preoperative chemotherapy in STS. Interestingly, MLPS was the only group in which the tailored chemotherapy regimen with trabectedin seems equally effective as the standard one [36]. That is why, in order to try to increase the survival of our patients with high grade MLPS, we are planning to conduct a new study with the 5 × 5 Gy RT regimen with delayed surgery used in this study. Local treatment will be combined with three cycles of chemotherapy with doxorubicin and ifosfamide, with the RT commencing immediately after the first cycle. From our other, not yet published, study dedicated to patients with marginally resectable STS (ClinicalTrials.gov Identifier: NCT03651375), where the aforementioned combined scheme of therapy is used, we know that the tolerance of this intensive scheme is good, with no evident overlapping toxicity [37].

The second issue requiring a possible adjustment is the fact that most patients (18/29 62%) in our group were treated with three-dimensional conformal RT that could have an impact on the rate of wound complications (37.9%). According to current guidelines, more conformal modern RT techniques such as VMAT (volumetric modulated arc therapy) or IMRT (intensity-modulated radiation therapy) could be considered to optimize the used treatment volume, improve dose conformity, and reduce toxicity to the surrounding tissue. Studies on IMRT in STS showed a reduced rate of toxicity (30.5% wound complication rate in the prospective trial by Sullivan et al. and 10.5% of late complication in the study by Wang et al.) and need for additional surgical procedures when compared to data coming from the randomized National Cancer Institute of Canada trial [6,38,39]. At the same time, the use of the IMRT technique allows the achievement of excellent local control rates [40]. A higher prevalence of modern RT techniques should be considered, especially when treating a patient with large tumors or located in unfavorable localization, for instance, the adductor compartment.

The third limitation of the study is its relatively small sample size and lack of comparator. However, due to the rarity of MLPS, conduction of randomized clinical trials is challenging or even impossible. Thus, other approaches should be considered, such as Bayesian trial design or data collection in prospective registries.

Moreover, the due decreased biologically equivalent dose (BED) of 5 × 5 Gy, if compared to 50 Gy in conventional fractionation, means that a definitively longer follow-up of our study is needed both for final efficacy and late toxicity. On the other hand, Haas et al. reviewed available data on efficacy of hypofractionated and conventionally fractionated RT regimens in STS. It was suggested that the dose–response relationship in preoperative RT exists for local control below 28 Gy in 8× 3.5 Gy. The conclusion was that the dose–response relationship between 28 Gy and 50 Gy may be marginal in preoperative RT [28]. Thus, 5 × 5 Gy may provide satisfactory BED for local control.

To conclude, our study can serve as a basis for modification of currently used combination therapy in patients diagnosed with MLPS. With preoperative 5 × 5 Gy delayed hypofractionated RT regimen, we have achieved the excellent local control rate. The primary endpoint of the study was met with a low rate of severe toxicity. The hypofractionated scheme is cost-effective and convenient to apply in every day clinical practice both for the patient and the treating team. Distant relapses occurred in patients with high-grade MLPS, implying the need for a more aggressive therapy in this group of patients, and an upcoming protocol is planned in our institution in order to address this need.

## Figures and Tables

**Figure 1 jcm-09-02471-f001:**
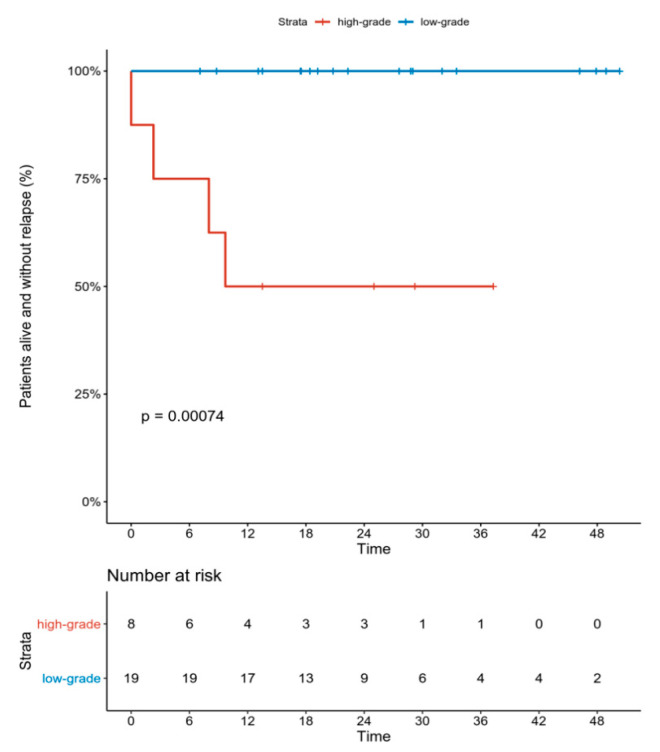
DFS (disease-free survival) according to tumor grade.

**Figure 2 jcm-09-02471-f002:**
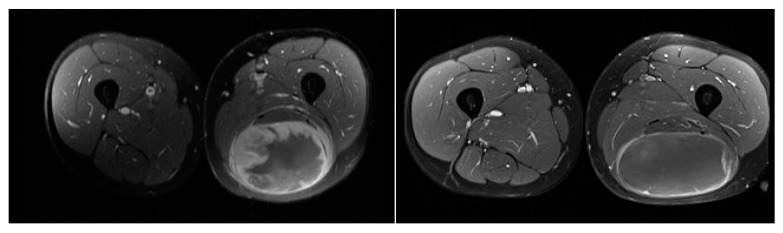
MRI images showing changes of the tumor and contrast uptake after treatment (**a**) prior to treatment, (**b**) prior to surgery.

**Figure 3 jcm-09-02471-f003:**
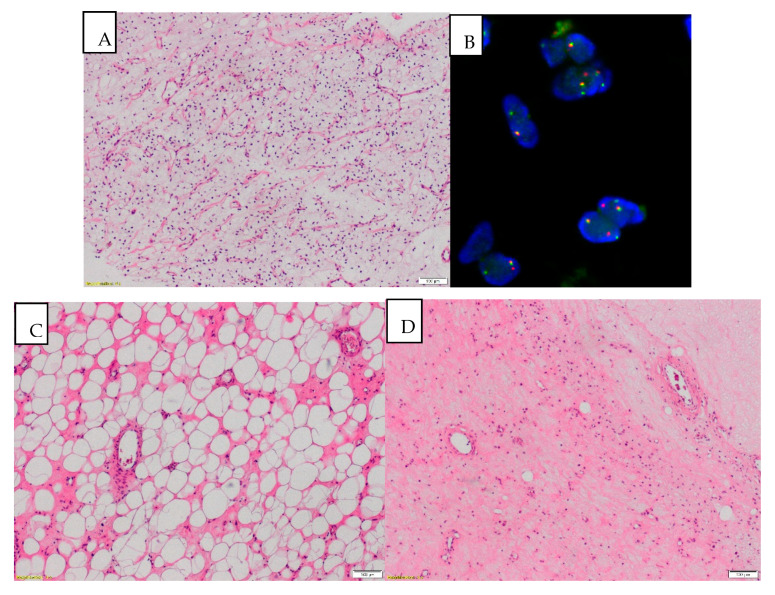
(**A**) MLPS (myxoid liposarcoma) before treatment: paucicellular, monomorphic, myxoid tumor with plexiform vasculature and scattered signet ring lipoblasts (magnification 40×). (**B**) Rearrangement of the DDIT3 gene detected on the FISH analysis. (**C**) After treatment: striking lipomatous differentiation induced by RTH can be a predominant pattern after treatment (magnification 40×), (**D**) After treatment: diminished cellularity, stromal fibrosis, reduced vasculature with thickened blood vessel walls and tumor infiltrating macrophages (TAM). Area of necrosis in the right upper corner (magnification 40×).

**Table 1 jcm-09-02471-t001:** Patient characteristics.

		*N* = 27 (%)
**Sex (%)**	female	13 (48)
	male	14 (52)
**Age (median [IQR])**		43 [32,58.5]
**Grade (%)**	high-grade	9 (33.3)
	low-grade	18 (66.6)
**Biopsy in NRI (%)**	no	7 (26)
	yes	20 (74)
**Size (median [IQR])**		13 [10,15]
**Size (%)**	T1	3 (11)
	T2	5 (18.5)
	T3	14 (52)
	T4	5 (18.5)
**Necrosis in biopsy (%)**	absent	20 (74)
	present	5 (18.5)
	unknown	2 (7.5)

NRI, National Research Institute of Oncology Warsaw Poland; IQR, interquartile range.

**Table 2 jcm-09-02471-t002:** Treatment toxicity, according to CTCAE (National Cancer Institute Common Terminology Criteria for Adverse Events) version 4.

*n* = 29(%)	All	Grade 1	Grade 2	Grade 3
**Radiation dermatitis**	12 (41.4)	10 (34.4)	1 (3.4)	1 (3.4)
**Acute toxicity** (≤3 months from surgery)	11 (37.9) *			
*Wound complication*		8 (29.6)	3 (11.1)	-
*Wound infection requiring systemic antibiotic*				5 (18.5)
**Late toxicity** (>3 months from surgery)	4 (13.8)			
*Lymphedema*		1 (3.4)	1 (3.4)	-
*Fibrosis*		1 (3.4)	1 (3.4)	-

* some different toxicities occurred in the same patient.

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
