# Peer review of "Hypofractionated Radiotherapy in Locally Advanced Myxoid Liposarcomas of Extremities or Trunk Wall: Results of a Single-Arm Prospective Clinical Trial"

_jcm, 2020, doi:10.3390/jcm9082471_

Round 1

Reviewer 1 Report

Becuase of characteristics of single arm study, confirmatory results are still needed.

Beside  the RECIST criteris, other response outcome is mandatory.

Especially, because all the patients underwent curative intent surgery, pathologic response (ex. tumor necrosis after radiotherapy) seems to be interesting and informative.  

Author Response

1. Because of characteristics of single arm study, confirmatory results are still needed.

         Thank you for this comment - of course, we fully agree. Unfortunately, it would be particularly challenging to carry out a randomized clinical trial in myxoid LPS comparing hypofractionated radiotherapy to standard fractionated one, mainly due to very high number of patients that would be required for such a study design. Still we plan to continue our research on hypofractionated radiotherapy in MLPS both combined with chemotherapy, as it is mentioned in the manuscript but also with the use of radiation alone on larger group of patients with more detailed assessment of response both radiological (response also by Choi criteria and based on diffusion-weighted MRI, according to the EORTC-STBSG recommendations) and pathological (induction of lipogenesis).

In the text we discussed the issue by adding (Discussion-Page 9): The third limitation of the study is its relatively small sample size and lack of comparator. However, due to rarity of MLPS, conduction of randomized clinical trials is challenging or even impossible. Thus, other approaches should be considered, such as Bayesian trial design or data collection in prospective registries.  

Moreover, the due decreased biologically equivalent dose (BED) of 5x5 Gy if compared to 50 Gy in conventional fractionation definitively longer follow-up of our study is needed both for final efficacy and late toxicity

2. Beside the RECIST criteria, other response outcome is mandatory.

 We agree with this suggestion. In this protocol we used the RECIST criteria for the assessment of response to treatment since we know that myxoid liposarcomas often respond in terms of size to radiotherapy. Of course it would be also interesting to see how the tumors respond to treatment also by the change of their density and to assess if this change has any impact on the treatment results (as it was assessed by the Italian group in 2018 doi: 10.1016/j.ejca.2018.01.071.) we plan to retrospectively asses those criteria in our patients group. In our future studies we plan on using both RECIST and Choi criteria.

In the text we added (Discussion-page 7):  The assessment of response to therapy has some shortcomings as we decided to use the RECIST criteria, which is still validated system for response evaluation in clinical trials. As it was aforementioned it has been shown that myxoid liposarcomas often respond to treatment with a significant size shrinkage, still we did not assess in a planned manner the changes in the density of the lesions. Choi criteria are used more commonly in the assessment of treatment in sarcomas[22] with some evidence that those criteria can be more adequate than RECIST in predicting the patient outcome[23]. We plan to retrospectively assess those criteria in our patients’ group and in our future studies we plan on using both RECIST and Choi criteria. 

3. Especially, because all the patients underwent curative intent surgery, pathologic response (ex. tumor necrosis after radiotherapy) seems to be interesting and informative. 

         Thank you for this remark, it is discussed on page 7. We performed a quite extensive assessment of pathological response to therapy. As it is mentioned in the text, in all analyzed surgical specimens, histological response features to treatment such as hyalinization, fibrosis, necrosis were detected. Also, in all but one patient adipogenesis induction was noted, as a response to therapy. None of these findings had impact on the patient’s outcome that is why we decided not to include more detailed pathological information and analyses in this text. We are preparing a separate paper focused only on pathological response features noted in patient with MPLS treated with preoperative radiotherapy

Reviewer 2 Report

Excellent job, congratulations to the authors.

Some minor comments:

Methods:

The authors included a substantial proportion of low grade tumors, which may have been managed with surgery alone according to current guidelines. They may include an explanation why these patients have been treated with neoadjuvant RT (for example because of threatened margins after multidisciplinary discussion as I suppose).

I would encourage the authors to have a look at the reponse rate according to Choi criteria also. Although MLPS are probably one of the best suited sarcoma subtypes to assess response by RECIST criteria because real shrinkage is seen in the majority of patients), it is also well known that Choi criteria are superior to RECIST in sarcomas in general. If possible one may think about adding response according to Choi, not instead but additionally to the mentioned RECIST based response rates.

Discussion:

The authors correctly explained the main advantages and concerns regarding hypofractionation. I feel that two points may be adjusted a little bit. One concern especially in sarcomas maybe be long-term efficacy because of decreased biologically equivalent dose (BED) if 5x5 if compared to 50 Gy in conventional fractionation given the alpha/beta values usually assumed for sarcomas. The authors may comment in one- or two sentences on that issue. Second I think the second part of statement „The main concern in the use of hypofractionation is a theoretically higher risk of late complications; however, it has not been shown in hypofractionated regimens in other cancers“ might be weakened a little bit. There are indeed reports on increased late toxicity with hypofractionation in some cancers (although depending on total dose as well). One should keep in mind that late toxicity plays a major role in the functional outcome of patients with extremity sarcomas and nearly all adjacent structures (muscle, bone, nerves) are late reacting tissues usually not ideal targets for hypofractionation. Moreover, the follow up seems to short to draw definitive conclusions on either efficacy nor late toxicity.

Author Response

1. The authors included a substantial proportion of low grade tumors, which may have been managed with surgery alone according to current guidelines. They may include an explanation why these patients have been treated with neoadjuvant RT (for example because of threatened margins after multidisciplinary discussion as I suppose).

 We added to the text an additional comment as indeed all the patients with low grade tumors were qualified to the protocol after a MDT discussion were it was decided that preoperative treatment is appropriate because of an extensive size of the lesion or its difficult location what created concern of the surgical margin.

 In the text we added (Page 2):  According to NCCN and ESMO guidelines, preoperative RT is not a routine treatment in patients with stage I sarcomas. However, it may be considered if we expect achieving oncologically not appropriate margins, especially in larger tumors (stage IB). Such a consideration was discussed during sarcoma multidisciplinary tumor board in each patient with locally advanced MLPS. We assumed a good response to preoperative RT that would lead to tumor downstaging as well as to elimination of microscopical infiltration of sarcoma cells around the primary tumor.

2. I would encourage the authors to have a look at the response rate according to Choi criteria also. Although MLPS are probably one of the best suited sarcoma subtypes to assess response by RECIST criteria because real shrinkage is seen in the majority of patients), it is also well known that Choi criteria are superior to RECIST in sarcomas in general. If possible one may think about adding response according to Choi, not instead but additionally to the mentioned RECIST based response rates.

 We fully agree with this suggestion. In this protocol we used the RECIST criteria for the assessment of response to treatment since we know that myxoid liposarcomas often respond in terms of size to radiotherapy. Of course it would be also interesting to see how the tumors respond to treatment also by the change of their density and to assess if this change has any impact on the treatment results (as it was assessed by the Italian group in 2018 doi: 10.1016/j.ejca.2018.01.071.) we plan on retrospectively asses those criteria in our patients group. In our future studies we plan on using both RECIST and Choi criteria.

In the text we added (Discussion-page 7):  The assessment of response to therapy has some shortcomings as we decided to use the RECIST criteria, which is still validated system for response evaluation in clinical trials. As it was aforementioned it has been shown that myxoid liposarcomas often respond to treatment with a significant size shrinkage, still we did not assess in a planned manner the changes in the density of the lesions. Choi criteria are used more commonly in the assessment of treatment in sarcomas[22] with some evidence that those criteria can be more adequate than RECIST in predicting the patient outcome[23]. We plan to retrospectively assess those criteria in our patients’ group and in our future studies we plan on using both RECIST and Choi criteria. 

3. The authors correctly explained the main advantages and concerns regarding hypofractionation. I feel that two points may be adjusted a little bit. One concern especially in sarcomas maybe be long-term efficacy because of decreased biologically equivalent dose (BED) if 5x5 if compared to 50 Gy in conventional fractionation given the alpha/beta values usually assumed for sarcomas. The authors may comment in one- or two sentences on that issue. Second I think the second part of statement „The main concern in the use of hypofractionation is a theoretically higher risk of late complications; however, it has not been shown in hypofractionated regimens in other cancers“ might be weakened a little bit. There are indeed reports on increased late toxicity with hypofractionation in some cancers (although depending on total dose as well). One should keep in mind that late toxicity plays a major role in the functional outcome of patients with extremity sarcomas and nearly all adjacent structures (muscle, bone, nerves) are late reacting tissues usually not ideal targets for hypofractionation. Moreover, the follow up seems to short to draw definitive conclusions on either efficacy nor late toxicity.

 Thank you very much for this valuable comment. As you mentioned of course it is too early for definite conclusions on long term safety and efficacy, still we have more data coming from our previous studies on the use of 5x5 scheme in preoperative treatment in soft tissue sarcomas (also in the myxoid liposarcomas patients). In our institution the 5x5 Gy scheme has been used for at least 15 years so the follow up is quite adequate. It is not fully published but, we do not see more late term toxicity when compared to data coming from the standard fractionation, also the efficacy is similar especially when it comes to patients with myxoid liposarcomas, which may be another argument in the discussion on changing the treatment regimen (lowering the total dose) in this specific group of patients.

 In the text we added the required changes on page 9. The third limitation of the study is its relatively small sample size and lack of comparator. However, due to rarity of MLPS, conduction of randomized clinical trials is challenging or even impossible. Thus, other approaches should be considered, such as Bayesian trial design or data collection in prospective registries.  Moreover, the due decreased biologically equivalent dose (BED) of 5x5 Gy if compared to 50 Gy in conventional fractionation definitively longer follow-up of our study is needed both for final efficacy and late toxicity. On the other hand, Haas et al. revieved available data on efficacy of hypofractionated and conventionally-fractionated RT regimens in STS. It was suggested that dose-response relationship in preoperative RT exists for local control below 28 Gy in 8x 3.5 Gy. The conclusion was that the dose-response relationship between 28 Gy and 50 Gy may be marginal in preoperative RT[28]. Thus, 5x5 Gy may provide satisfactory BED for local control.